# Forensic Application of Stable Isotopes to Distinguish between Wild and Captive Turtles

**DOI:** 10.3390/biology11121728

**Published:** 2022-11-29

**Authors:** John B. Hopkins, Cheryl A. Frederick, Derek Yorks, Erik Pollock, Matthew W. H. Chatfield

**Affiliations:** 1Center for Wildlife Studies, 36A High St., Camden, ME 04843, USA; 2Maine Department of Inland Fisheries and Wildlife, Bangor, ME 04441, USA; 3Stable Isotope Laboratory, University of Arkansas, Fayetteville, AR 72701, USA; 4School of Biology and Ecology, University of Maine, Orono, ME 04469, USA

**Keywords:** stable isotopes, *δ*^13^C, *δ*^15^N, carbon, nitrogen, forensics, wood turtles

## Abstract

**Simple Summary:**

Wildlife trafficking is a major contributor to global biodiversity loss, especially reptiles, which are confiscated by law enforcement more than any other vertebrate class. Wildlife forensic experts can use chemicals from animal tissues to determine the origin of confiscated animals. Such physical evidence can help law enforcement prosecute wildlife traffickers in court and hold poachers accountable. In this study, we developed a statistical tool that can be used to determine if a confiscated wood turtle (*Glyptemys insculpta*) from Maine came from the wild or captivity. We used carbon and nitrogen stable isotopes from wood turtle claw tips to construct a statistical model that correctly classified all wild turtles as wild and nearly all captive turtles as captive (predictive accuracy 97.2%). Results from our study can be used to assist law enforcement in Maine and to develop a forensics tool used to help combat the illegal turtle trade.

**Abstract:**

Wildlife traffickers often claim that confiscated animals were captive-bred rather than wild-caught to launder wild animals and escape prosecution. We used stable isotopes (*δ*^13^C and *δ*^15^N) derived from the claw tips of wild wood turtles from Maine and captive wood turtles throughout the eastern U.S. to develop a predictive model used to classify confiscated wood turtles as wild or captive. We found that the claw tips of wild and captive wood turtles (*Glyptemys insculpta*) were isotopically distinct. Captive turtles had significantly higher *δ*^13^C and *δ*^15^N values than wild turtles. Our model correctly classified all wild turtles as wild (100%) and nearly all captive turtles as captive (94%). All but two of the 71 turtles tested were successfully predicted as wild or captive (97.2% accuracy), yielding a misclassification rate of 2.8%. In addition to our model being useful to law enforcement in Maine, we aim to develop a multi-species model to assist conservation law enforcement efforts to curb illegal turtle trafficking from locations across the eastern United States and Canada.

## 1. Introduction

The illegal pet trade often involves “laundering” animals to hide their origins, creating legal loopholes that include passing an animal off as captive-bred [1,2]. One way to combat such wildlife laundering is to develop new forensic tools for determining the origins of animals seized by law enforcement [3], including predictive models used to classify a confiscated animal as poached from the wild versus raised in captivity at a private residence or commercial breeding facility. Results from such a quantitative analysis could be presented in court as evidence for prosecuting alleged wildlife traffickers [4].

Stable isotopes analysis (SIA) can be used as a powerful forensics tool for combating the illegal wildlife trade [5]. SIA of animal tissues can be used to differentiate between wild-caught and captive animals based on isotopic differences in the composition of their tissues, which reflect differences in the diet and environments they experienced during tissue growth [6]. Animals raised in captivity often have routine diets that are high in nutrients and are generally less varied than those consumed by wild animals [7,8]. Differences in diet can cause isotopic dissimilarity between groups, allowing researchers to differentiate between wild and captive animals, such as American mink (*Mustela vision*; [9]), Atlantic salmon (*Salmo salar*; [10]), Burmese pythons (*Python bivittatus*; [5]), and wood turtles (*Glyptemys insculpta*; this study). 

Despite their overwhelming prevalence in the legal and illegal wildlife trade (much more so than mammals, birds, fish, invertebrates, and amphibians combined), reptiles, including freshwater turtles, have received little conservation funding or meaningful law enforcement attention, leading to population declines [11]. For instance, the collection of wood turtles is a contributor to population losses throughout their range, as evidenced by confiscations (see [12]). As a result of their declining numbers in the U.S. and Canada, wood turtles are currently listed as Endangered by the International Union for Conservation of Nature (IUCN) and have been proposed for listing under the U.S. Endangered Species Act [13]. 

In this study, we explored the use of carbon and nitrogen stable isotope ratios (expressed as *δ*^13^C and *δ*^15^N, respectively) derived from the claw tips of wild wood turtles from Maine and captive turtles collected from a variety of facilities throughout the eastern U.S. to (i) test predictions deduced from the wild- versus captive-feeding hypothesis, and (ii) develop a predictive model used to determine if a confiscated wood turtle from a given region, in this case, Maine, has wild or captive origins. Such information would help prevent wildlife trafficking of wood turtles in Maine and inform future strategies to expand our work to other geographic regions and species of concern. 

## 2. Materials and Methods

### 2.1. Sampling

Wild wood turtles are considered riparian specialists, moving seasonally between aquatic and upland habitats, yet are omnivorous dietary generalists [14]. Wood turtles naturally concentrate in streams or stream-adjacent habitats in the spring and fall (reviewed in [15]) when turtles are located and routinely handled for population surveys and ecological studies. In 2020 and 2021, with the aid of our partners in the field, we collected claw tips from 35 wild wood turtles sampled from three areas in Maine during either the spring or fall (Table 1). Due to poaching concerns, details about the locations of wild turtles are not included here.

During the same years, we also obtained claw tips from 36 captive wood turtles from a network of 12 facilities throughout the eastern United States. Captive turtles were also sampled in the spring and fall, although dates were more variable than wild turtles (Table 1). 

We collected samples using minimally invasive methods described in Hopkins et al. [16]. Specifically, participants used cat nail trimmers to clip off 1–3 mm of claw tip from one toe from each of two different feet. Participants stored all claw tips in paper envelopes, plastic bags, or plastic vials, labeling each with the date, species, age class, sex, and a unique identification number. Following Aresco et al. [17], who found turtle claw tissue reflected *δ*^15^N uptake in <6 months and *δ*^13^C uptake at >6 months, our wild samples were only taken from adults (ensuring well over one year of growth). Similarly, all sampled captive animals had been held for at least one year to allow ample time for any dietary changes from a wild to captive environment to be reflected in their claw tips [17].

### 2.2. Stable Isotope Analysis

We conducted stable isotope analysis at the University of Arkansas Stable Isotope Lab (UASIL). Staff at the UASIL weighed ~0.3 mg of claw material using a microbalance (Sartorius SC-2); wrapped each weighed sample in a tin capsule; and analyzed all samples using an EA-Isolink elemental analyzer interfaced via ConFlo IV to a Delta V plus isotope ratio mass spectrometer (Thermo Electron, Bremen, Germany). UASIL ran the system in a dual column mode for combustion (1020C), reduction (620C), and flow (100 mL/min), and CO_2_ and N_2_ gases were separated on 1/8” 0.5M GC column at 33C (proprietary phase Thermo Fisher Scientific, Waltham, MA, USA). The system was equipped with a Costech zero blank autosampler with a 49-position carousel; each full run consisted of 18 standards, 1 blank, and 30 samples. Staff at the UASIL normalized raw instrument delta values to international scale values using standards USGS 41a (*n* = 19) and USGS 8573 (*n* = 19) with *δ*^13^C = 36.55, −26.39 and *δ*^15^N = 47.55, −4.52, respectively. Standard reproducibility varied between 0.07 and 0.09 per mil (‰). Replicate standards had reproducibility better than 0.1‰ for both carbon and nitrogen.

### 2.3. Statistical Comparisons

Although we were primarily interested in comparing stable isotope values for wild and captive turtles, we also explored the possible differences in sex-based life history strategies and our seasonal sampling regimes [18]. Before performing the statistical comparisons described below, we first assessed the normality and homoscedasticity of *δ*^13^C and *δ*^15^N values for each of the three groups using Shapiro–Wilk and Levene’s tests, respectively. We also used Levene’s test to test the prediction that the variance of *δ*^13^C and *δ*^15^N values for captive turtles would be lower than wild turtles (our prediction was deduced from the hypothesis that captive turtle diets are less varied than their wild counterparts). We compared stable isotope values for captive and wild turtles, males and females, and turtles sampled in the spring and fall using *t*-tests, ANOVA, and Tukey tests if the stable isotope values for groups were normally distributed and had equal variance, or Mann–Whitney, Kruskal–Wallis, and Dunn’s (with Bonferroni correction) tests, if the stable isotope values for groups were non-parametric and/or they had non-constant variance. We conducted all analyses in R [19] using α=0.05.

### 2.4. Predictive Model

We used the package glmnet in R to fit generalized logistic regression models via penalized maximum likelihood. We also performed K-fold cross-validation using the glmnet package to select the best model from all combinations of models that include *δ*^13^C, *δ*^15^N, and sex as predictors. We did not include season as a variable in our candidate set because, in most cases, captive turtles did not have a seasonal diet. We reported the sensitivity (true positive rate), specificity (true negative rate), accuracy (success rate), misclassification rate (incorrect classification rate), optimal decision threshold, and the area under the receiver operating characteristic (AUROC) curve for our top model.

## 3. Results

### 3.1. Statistical Comparisons

All groups used in our statistical tests were normally distributed, except for wild turtles *δ*^13^C values (*W* = 0.926, *p* = 0.0213). We also found that the variance of *δ*^13^C values for turtles by sex (*F* = 8.745, *p* < 0.005) and season (*F* = 10.755, *p* < 0.005) were non-constant (Figure 1). Our data did not match the prediction that the variance of *δ*^13^C values for captive turtles would be lower than wild turtles; instead, we found the variance of *δ*^13^C values were greater for captive turtles (s^2^ = 1.64) than wild turtles (s^2^ = 0.12) and *δ*^15^N values for captive turtles (s^2^ = 1.42) were not significantly different from wild turtles (s^2^ = 1.67) (Figure 1, Table 1). We conducted non-parametric tests when comparing *δ*^13^C values for turtles and parametric tests when comparing their *δ*^15^N values. We learned that captive turtles were heavier in ^13^C and ^15^N than wild turtles, as indicated by their elevated *δ*^13^C (captive: −21.3 ± 1.3; wild: −24.6 ± 0.35; *W* = 1252, *p* < 0.005) and *δ*^15^N (captive: 9.0 ± 1.2; wild: 6.2 ± 1.3; *t* = 9.2838, *df* = 68.199, *p* < 0.005) values (Figure 1); this was not the case, however, for captive females versus males, wild females versus males, captive turtles sampled in the spring versus fall, and wild turtles sampled in the spring versus fall (Appendix A).

### 3.2. Predictive Model

Using an optimal discrimination threshold value of 0.48, our top model, which included all predictors (Appendix A), classified all 35 wild wood turtles as wild (100% sensitivity) and 34 of 36 captive turtles as captive (94% specificity) (Table 1). The two captive turtles predicted as wild had the lowest *δ*^13^C values for captive turtles and lower than average *δ*^15^N values. The model coefficients suggest that the most important predictor was *δ*^13^C, followed by being female, *δ*^15^N, and being male (Appendix A). Overall, our top model was 97.2% accurate at correctly classifying Maine wood turtles as wild or captive with a misclassification rate of 2.8%. A 0.997 area under the ROC curve suggests a very high predictive capacity for correctly classifying turtles as wild or captive.

## 4. Discussion

Our examination of stable isotopes derived from wood turtle claw tips yielded two principal findings. First, and most importantly, we found that wild wood turtles from Maine and captive wood turtles from a variety of facilities were isotopically distinct (Figure 1). Captive turtles had significantly higher *δ*^13^C and *δ*^15^N values; as a result, our predictive model correctly classified all wild turtles as wild and had a very low overall misclassification rate (<3%). Second, we learned that our data did not match the prediction that the stable isotope values for wild turtles would vary more than captive animals; instead, we found that wild turtles had lower isotopic variance than captive animals, suggesting wild wood turtle diets were less diverse. 

Captive turtles likely had greater *δ*^13^C and *δ*^15^N values than wild turtles because their formulated diet contained corn and animal protein. Past studies found that the tissues of animals that forage for foods that are anthropogenic in origin, including those foods high in corn and animal protein, have elevated *δ*^13^C and *δ*^15^N values, respectively. For instance, hair from American black bears (*Ursus americanus*) that feed on human foods had greater mean *δ*^13^C values (<0.6‰) and *δ*^15^N values (2‰) than those on a wild diet [20]; claws from farm-raised American mink were, on average, far greater (4‰) than their wild counterparts [9]; muscle from farmed Atlantic salmon had greater mean *δ*^13^C values (>1‰) than wild salmon sampled in Newfoundland [10]; and skin from captive Burmese pythons (2‰) had greater mean *δ*^13^C values than skin from wild pythons [5]. 

Unlike previous studies that show greater variation in stable isotopes for wild animals than their captive counterparts [5,9,10], we did not find support for the hypothesis that wild wood turtles have more diverse diets than captive turtles. We learned that among the captive facilities that contributed samples to this study, no two fed their wood turtles the same diet. They varied in the use and type of their commercial diet, mixes of fruits and vegetables, and animal protein sources. Unfortunately, we did not have large enough sample sizes to test the prediction as it applies to each captive facility. If we had, we might have found that although turtles are isotopically diverse among facilities, they are not within each facility. Conversely, it is possible that temporal, spatial, or other unknown factors constrain dietary diversity in wild turtles to a greater extent than anticipated given their complex natural history. 

Wood turtle ecology provides context for two other isotopic patterns we observed in our sample of wild turtles. First, while there are some variations in seasonal habitat use and movement patterns between males and females [21], we saw no isotopic differences between sexes, suggesting that minor divergences in life history strategies do not result in significant dietary differences (Appendix A). Second, we found that the isotopic composition of turtle claw tips collected in spring versus fall did not differ, which could have resulted from a long dormancy period (and lack of growth) of turtles as they overwinter in their aquatic hibernacula [15] (Appendix A). 

With over a 97% accuracy rate, our model correctly identified all wild Maine wood turtles as wild and all but two captive turtles as captive. Although there is a small chance (<3%) that captive turtles in Maine could be predicted as wild by our current model, our data inform and validate this approach. Since captive diets were highly variable, reflected in isotopic variability, it is not surprising that two of these turtles were outliers. Differences in diets offered, food preferences, and activity (e.g., one turtle lost two limbs before taken into captivity, which may have influenced wear on claw tips) are likely explanations for their greater resemblance to wild turtles. We believe that with greater samples sizes, examination of confiscated turtles, and further model refinements, isotopic profiles may be used to discriminate wild versus captive turtles with 100% accuracy.

## 5. Conclusions

To our knowledge, there are no other studies that used stable isotopes to distinguish between wild and captive freshwater turtles. We view the use of *δ*^13^C and *δ*^15^N isotope values as the first crucial step in developing a wildlife forensic tool used to help combat the illegal turtle trade by assisting conservation law enforcement efforts in the courtroom. Although our model is nearly perfect at classifying wood turtles in Maine as wild or captive, our goal is to develop a more general model that extends beyond Maine with a predictive accuracy of 100%. In the future, we hope to improve the predictive capacity of our model by including wood turtles across their geographic range, other turtle species of conservation concern, and additional predictors, including other stable isotopes (e.g., *δ*^2^H and *δ*^18^O) and chemical tracers. 

## Figures and Tables

**Figure 1 biology-11-01728-f001:**
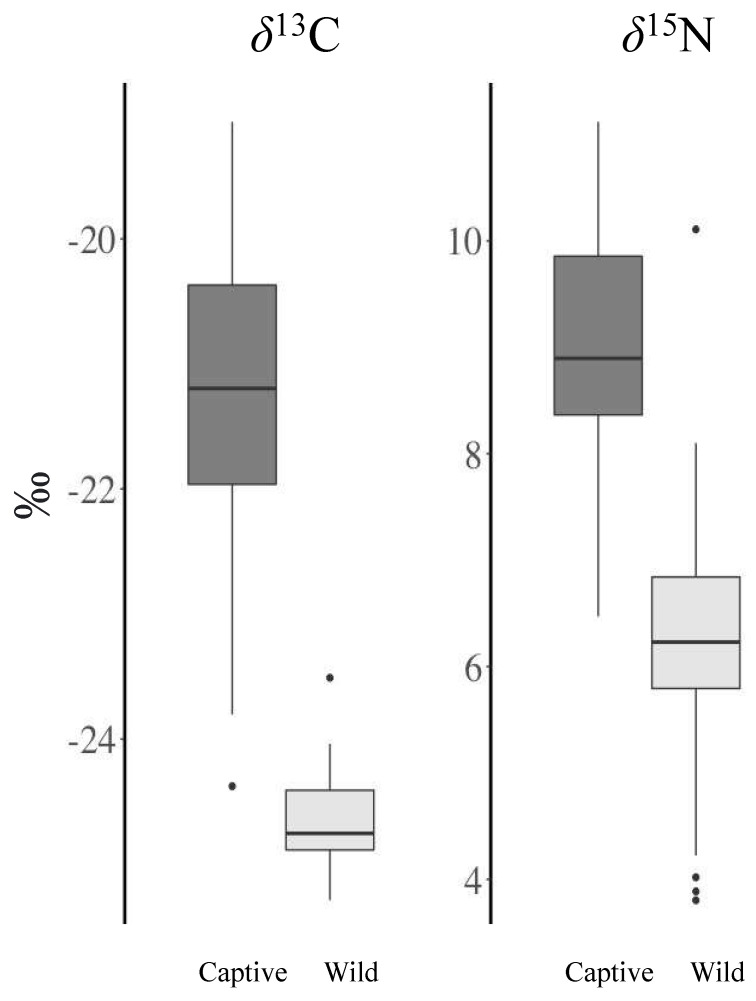
Carbon (*δ*^13^C) and nitrogen (*δ*^15^N) stable isotope values (per mil, ‰, measured by IRMS) derived from the claw tips of wild wood turtles captured in Maine and captive wood turtles sampled at various animal care facilities throughout the eastern United States (Table 1).

**Table 1 biology-11-01728-t001:** Carbon (*δ*^13^C) and nitrogen (*δ*^15^N) stable isotope values for wild wood turtles captured in Maine and captive wood turtles sampled at various animal care facilities throughout the eastern United States. Prob Wild is the predicted probability that each turtle in the study is wild based on the stable isotopic composition of their claw tips and their sex.

Class	Location	Sex	Season	*δ*^13^C	*δ*^15^N	Prob Wild
Captive	ACF1	F	Fall	−21.07	9.98	0.0002
Captive	ACF2	F	Fall	−22.61	8.73	0.0226
Captive	ACF3	F	Fall	−22.98	8.59	0.0610
Captive	ACF3	F	Fall	−20.27	8.62	0.0001
Captive	ACF3	F	Fall	−19.14	9.90	0.0000
Captive	ACF4	F	Fall	−20.26	9.47	0.0000
Captive	ACF4	F	Fall	−20.91	9.11	0.0002
Captive	ACF4	F	Fall	−20.48	8.26	0.0002
Captive	ACF4	F	Fall	−24.38	8.40	0.7187
Captive	ACF4	F	Fall	−23.62	7.47	0.4788
Captive	ACF4	F	Fall	−19.98	9.20	0.0000
Captive	ACF2	F	Spring	−19.55	10.45	0.0000
Captive	ACF3	F	Spring	−20.96	10.46	0.0001
Captive	ACF4	F	Spring	−19.61	10.84	0.0000
Captive	ACF5	F	Spring	−21.92	10.05	0.0012
Captive	ACF6	F	Spring	−21.10	8.97	0.0004
Captive	ACF7	F	Spring	−21.87	11.12	0.0004
Captive	ACF7	F	Spring	−21.38	10.92	0.0001
Captive	ACF8	F	Spring	−20.40	9.48	0.0000
Captive	ACF9	F	Spring	−21.25	8.79	0.0007
Captive	ACF9	F	Spring	−21.14	8.79	0.0005
Captive	ACF9	F	Spring	−20.46	8.97	0.0001
Captive	ACF9	F	Spring	−21.83	8.74	0.0032
Captive	ACF9	F	Spring	−22.54	6.80	0.1031
Captive	ACF9	F	Spring	−22.64	6.50	0.1632
Captive	ACF9	F	Spring	−22.40	7.05	0.0612
Captive	ACF9	F	Spring	−21.46	9.48	0.0006
Captive	ACF9	M	Fall	−21.26	8.82	0.0023
Captive	ACF10	M	Fall	−20.27	9.84	0.0001
Captive	ACF11	M	Fall	−23.80	8.12	0.7136
Captive	ACF4	M	Spring	−21.55	8.12	0.0090
Captive	ACF11	M	Spring	−19.89	8.41	0.0001
Captive	ACF11	M	Spring	−19.06	8.18	0.0000
Captive	ACF11	M	Spring	−21.82	10.34	0.0022
Captive	ACF12	M	Spring	−21.01	9.51	0.0006
Captive	ACF12	M	Spring	−22.08	6.47	0.1368
Wild	FIELD1	F	Fall	−24.82	4.22	0.9973
Wild	FIELD1	F	Fall	−24.90	5.39	0.9936
Wild	FIELD1	F	Fall	−24.75	6.23	0.9802
Wild	FIELD1	F	Fall	−24.89	3.89	0.9984
Wild	FIELD1	F	Fall	−24.04	7.31	0.7521
Wild	FIELD2	F	Fall	−24.61	4.02	0.9963
Wild	FIELD2	F	Fall	−24.57	6.47	0.9613
Wild	FIELD1	F	Spring	−24.20	8.10	0.6866
Wild	FIELD1	F	Spring	−24.71	5.75	0.9858
Wild	FIELD1	F	Spring	−24.84	5.84	0.9889
Wild	FIELD1	F	Spring	−24.60	6.63	0.9582
Wild	FIELD1	F	Spring	−24.78	6.23	0.9812
Wild	FIELD1	F	Spring	−25.29	10.11	0.8359
Wild	FIELD1	F	Spring	−25.00	4.37	0.9981
Wild	FIELD1	F	Spring	−25.00	3.80	0.9989
Wild	FIELD1	F	Spring	−24.30	7.69	0.8047
Wild	FIELD1	F	Spring	−24.88	7.56	0.9518
Wild	FIELD1	F	Spring	−24.83	6.15	0.9847
Wild	FIELD1	F	Spring	−24.61	4.57	0.9939
Wild	FIELD1	F	Spring	−24.90	6.40	0.9837
Wild	FIELD1	F	Spring	−24.18	6.45	0.9063
Wild	FIELD1	M	Fall	−24.15	6.16	0.9738
Wild	FIELD1	M	Fall	−23.51	7.41	0.6998
Wild	FIELD1	M	Fall	−24.93	6.39	0.9952
Wild	FIELD1	M	Fall	−24.38	6.69	0.9757
Wild	FIELD1	M	Spring	−24.76	6.49	0.9921
Wild	FIELD1	M	Spring	−24.43	6.83	0.9756
Wild	FIELD1	M	Spring	−24.87	5.91	0.9964
Wild	FIELD1	M	Spring	−24.43	7.38	0.9600
Wild	FIELD1	M	Spring	−24.39	7.15	0.9637
Wild	FIELD3	M	Spring	−25.02	5.93	0.9975
Wild	FIELD3	M	Spring	−24.89	6.03	0.9962
Wild	FIELD3	M	Spring	−24.77	5.91	0.9954
Wild	FIELD3	M	Spring	−24.39	6.85	0.9722
Wild	FIELD3	M	Spring	−24.68	5.63	0.9956

## Data Availability

Not applicable.

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
