# Peer review of "Forensic Application of Stable Isotopes to Distinguish between Wild and Captive Turtles"

_biology, 2022, doi:10.3390/biology11121728_

Round 1
Reviewer 1 Report
The manuscript titled “Forensic application of stable isotopes to distinguish between 2 wild and captive turtles” describes the application of stable isotopes to reduce wildlife tracking and mitigate biodiversity loss. The well-written manuscript provides novel research that highlights the potential use of stable isotopes to distinguish between wild and captive turtles from differences in diet to assist law enforcement. I think the manuscript is suitable for publication, but I suggest the authors consider my suggestions embedded within the pdf.

Reviewer 2 Report
The authors conducted very interesting studies using stable isotope analysis to distinguish wild and captive turtles. It is a good application of stable isotope analysis. The major thing I would suggest improving is to add detailed description of methods, including instrument, column, analysis temperature on EA and IRMS (isotope ratio mass spectrometry), and sample preparation for each run.
1. The description for experimental section of stable isotope analysis is not sufficient. Please address the instrument information and method, such as temperature, column, and flow rate.
2. Please describe how to prepare the samples for sample analysis on the instrument and how much for each run.
3. Normally, the δ13C value of the samples should be determined with a standard deviation ≤ 0.5‰. Please check if the data analysis is within this range.
4. It looks like the isotope ratio is greater in captive turtles or some other species than wild ones due to a high-quality animal diet. Does that mean the wild animals’ diet are lack of nutrients?
5. Besides diet, could be other factors involved in the data difference, such as activity frequency?
6. Did the authors perform duplicates or triplicates analysis for the precision purpose?
